# Inflammatory Transformation of Skin Basal Cells as a Key Driver of Cutaneous Aging

**DOI:** 10.3390/ijms26062617

**Published:** 2025-03-14

**Authors:** Shupeng Liu, Sheng Lu, Zhiping Pang, Jiacheng Li, Meijuan Zhou, Zhenhua Ding, Zhijun Feng

**Affiliations:** Department of Radiation Medicine, Guangdong Provincial Key Laboratory of Tropical Disease Research, School of Public Health, Southern Medical University, Guangzhou 510515, China; 20220062@smu.edu.cn (S.L.); ls804469506@163.com (S.L.); zhiping_pang@163.com (Z.P.); jacky8715@126.com (J.L.); lkzmj@smu.edu.cn (M.Z.)

**Keywords:** skin aging, basal cell transformation, inflammatory trajectory, IFI27^+^ cells, single-cell transcriptomics, GWAS

## Abstract

This study comprehensively investigated keratinocyte subpopulation heterogeneity and developmental trajectories during skin aging using single-cell sequencing, transcriptomics, and facial aging-related genome-wide association studies (GWAS) data. We identified three major subpopulations: basal cells (BCs), spinous cells (SCs), and IFI27^+^ keratinocytes. Single-cell pseudotime analysis revealed that basal cells can differentiate along two distinct paths: toward spinous differentiation or the inflammatory state. With aging, the proportion of IFI27^+^ cells significantly increased, displaying more active inflammatory and immunomodulatory signals. Through cell–cell communication analysis, we found that the signaling pathways, including NOTCH, PTPR, and PERIOSTIN, exhibited distinct characteristics along different branches. Integration of the GWAS data revealed significant loci on chromosomes 2, 3, 6, and 9 that were spatially correlated with key biological pathways (including antigen processing, oxidative stress, and apoptosis). These findings reveal the complex cellular and molecular mechanisms underlying skin aging, offering potential targets for novel diagnostic approaches and therapeutic interventions.

## 1. Introduction

The skin is the largest organ in the human body and the most direct reflection of the aging process [1,2]. Anatomically, the skin primarily consists of the epidermis and dermis, connected by the basement membrane, which together maintain the skin’s barrier function, immune defense, and aesthetic properties [3,4]. In the epidermis, keratinocytes—which constitute the majority of cells—continuously renew, differentiate, and ultimately form the stratum corneum, providing crucial protection against the external environment. However, with advancing age, the proliferation, differentiation, and repair capabilities of keratinocytes gradually decline, dermal matrix remodeling becomes imbalanced, and the barrier function weakens, leading to typical aging phenotypes such as dryness, laxity, wrinkles, and pigmentation [5,6,7,8].

Recent studies have shown that skin aging is not merely the functional decline in a single cell type but rather the result of complex interactions among multiple cell populations [9,10,11] (including keratinocytes, fibroblasts, immune cells, etc.). In young skin, the epidermis and dermis maintain dynamic coordination through various cytokines and signaling pathways (such as TGF-β, Wnt/β-catenin, Notch, etc.), maintaining efficient extracellular matrix renewal and antioxidant defense capabilities, ensuring barrier integrity and repair efficiency. Once this network is affected by both intrinsic genetic factors and external environmental factors (UV radiation, pollution, stress, etc.), it can trigger cellular signaling disorders, exacerbate the aging phenotype of keratinocytes, decrease dermal fibroblast activity, and activate chronic inflammatory pathways, leading to or accelerating skin dysfunction [12,13,14,15]. However, the current research on “keratinocyte dynamics” and “cell–cell signaling” during skin aging remains insufficient. On one hand, the proliferation, differentiation, and apoptosis of keratinocytes show staged and spatial changes under multiple signal regulations but their dynamic patterns under different aging stages and external stimuli have not been systematically elucidated. On the other hand, there is a lack of in-depth research on the signaling interaction mechanisms between fibroblasts, immune cells, neuroendocrine cells, and keratinocytes, particularly in understanding the spatiotemporal distribution and key regulatory sites of various secretory factors and receptor ligands under different aging contexts. Therefore, it is necessary to utilize multi-level approaches combining cell biology, molecular biology, and omics to deeply analyze the key nodes and pathways of keratinocyte dynamics and cell–cell signaling during skin aging [16,17,18,19,20].

Based on this, our study obtained single-cell RNA-sequencing (scRNA-seq) data of aged and young skin from public databases, focusing on extracting the keratinocyte components and identifying different keratinocyte subtypes using dimensionality reduction and clustering methods. Subsequently, through trajectory analysis and cell–cell communication analysis, we systematically compared the differences between keratinocyte subgroups in aged and young groups. From a functional perspective, we further elucidated the changes occurring in various subgroups during biological processes such as cell proliferation, differentiation, and apoptosis, and cross-validated whether related genetic variations affect the expression of skin aging-related genes through integration with genome-wide association studies (GWAS) data. This study not only provides new insights into understanding keratinocyte dynamics and their cellular signaling interaction networks during skin aging but also lays an important theoretical foundation for future early diagnosis of skin aging, biomarker screening, and the development of personalized anti-aging intervention strategies.

## 2. Results

### 2.1. Keratinocyte Subpopulations and Developmental Trajectories in Young and Aged Skin

Overall, single-cell RNA-sequencing (scRNA-seq) analysis revealed three prominent cell clusters (clusters 0–2) within keratinocytes from both young and aged skin (Figure 1A). According to typical keratinocyte marker genes (Figure 1B), cluster 0 predominantly expresses the spinous cell (SC) markers (*KRT1* and *KRT10*), whereas cluster 1 is characterized by high expression of the basal cell (BC) markers (*KRT5* and *KRT14*). Notably, cluster 2 co-expresses markers from both SCs and BCs, and also exhibits elevated levels of *IFI27* and *MMP2* (Figure 1C). Based on these marker profiles, we categorized the keratinocytes into three main groups: BCs, SCs, and an IFI27^+^ keratinocyte population (Figure 1D). Compared with young skin, aged skin showed a decrease in BCs but a significant increase in both SCs and IFI27^+^ keratinocytes (Figure 1E), suggesting that IFI27^+^ keratinocytes may play an important role in the skin aging process.

To further elucidate the developmental trajectories of skin keratinocytes in young and aged individuals, we performed pseudotime analysis using “Monocle 2”. The results revealed two clearly diverging developmental paths (Figure 1F). Based on the developmental nodes identified in the trajectory, the keratinocytes were categorized into three distinct states (Figure 1G). We defined the initial stage as the early state (ES), the upward-right branch as advanced state direction 1 (AS1), and the downward-left branch as advanced state direction 2 (AS2) (Figure 1G). In the ES stage, cells are primarily BCs, whereas the AS1 branch is dominated by SCs, and the AS2 branch is mainly composed of IFI27^+^ keratinocytes (Figure 1H). Notably, a comparison of aged and young samples indicated that the IFI27^+^ epithelial subpopulation (AS2) is more prevalent in aged skin (Figure 1H). When examining the proportion of each cell subpopulation in each stage, BCs predominate in the ES, SCs dominate the AS1 (with a smaller fraction of BCs), and the AS2 is largely made up of IFI27^+^ keratinocytes (with some BCs; Figure 1I). These findings provide a mechanistic framework suggesting that BCs continue to serve as the principal progenitor pool at early stages in both young and aged skin yet they bifurcate into SC^−^ and IFI27^+^ dominant lineages as aging progresses—a process that may underlie divergent regenerative and inflammatory trajectories contributing to age-associated skin changes.

We next analyzed the gene expression patterns characteristic of each developmental state (Figure 1J). We found that ES cells primarily express genes related to maintaining keratinocyte activity, such as *AREG* and *KRT15*. In contrast, the AS1 branch is marked by high levels of the spinous-layer keratins *KRT1* and *KRT10*. The AS2 branch shows a distinct increase in inflammatory-related genes (*CD74*, *CTS3*, and *S100A4*) and elevated expression of mesenchymal markers (e.g., *VIM*). The top ten marker genes for each state are displayed in Figure 1K. Further enrichment analysis confirmed that ES cells are primarily associated with biological processes involved in keratinocyte differentiation, as well as IL-17 and TNF signaling. AS1 cells are enriched in processes related to carbon dioxide and oxygen transport, as well as the estrogen signaling pathway. Meanwhile, the IFI27^+^ epithelial subgroup (AS2) is strongly linked to lymphocyte activation and antigen-processing/presentation pathways. Combining these findings with our trajectory analysis, we propose that, as skin ages, BCs may adopt a pro-inflammatory phenotype—exemplified by IFI27^+^ keratinocytes—thereby fostering a heightened inflammatory milieu that not only disrupts the homeostatic renewal of the epidermis but also accelerates age-associated functional decline in the skin.

### 2.2. Keratinocyte Subpopulation Interactions in Young and Aged Skin

Because BCs serve as the principal progenitor population at the onset of development, alterations in their signaling may impact the differentiation of all keratinocytes. To better elucidate the interplay among subpopulations at different developmental states, we subdivided BCs into three stages based on a pseudotime trajectory analysis: BC-ES (early-state BCs), BC-AS1 (BCs inclined toward the spinous-cell lineage), and BC-AS2 (BCs inclined toward the IFI27^+^ lineage). This approach allows us to capture how cell–cell interaction patterns shift as BC-ES cells diverge into distinct developmental pathways.

From the perspective of cell–cell contacts (Figure 2A,B), varying degrees of connectivity exist among all epithelial subpopulations. The IFI27^+^ subpopulation shows a relatively high number of connections with other groups (Figure 2A). However, in terms of signal strength, SC cells exhibit the most robust internal interactions (Figure 2B). A similar pattern emerges for ECM-receptor interactions (Figure 2C,D): the IFI27^+^ subgroup has extensive interactions with other populations (Figure 2C), with particularly strong ECM-receptor signals observed between IFI27^+^ cells and both BC-ES and SCs (Figure 2D). With respect to secretory signals (Figure 2E,F), each epithelial subpopulation demonstrates distinct levels of secretory output. Notably, IFI27^+^ cells, BC-AS1, and BC-AS2 produce a comparatively large number of secreted factors (Figure 2E). Again, in terms of interaction strength, the IFI27^+^ population displays especially prominent secretory signaling with BC-ES and SCs (Figure 2F).

Based on these overarching observations of cell–cell contacts, ECM-receptor dynamics, and secretory signals, we next examined the specific molecular features that distinguish each mode of interaction in more detail. For cell–cell contacts (Figure 2G), the IFI27^+^ subpopulation exhibits a marked increase in both outgoing and incoming signals, including immune-related factors (e.g., MHC, CD99, CD40, and CD86), consistent with a pro-inflammatory role. The BC-ES subpopulation shows enhanced output of APP, NOTCH, SIRP, and GP1BA, along with a strengthened reception of signals such as CDH1 (from SC), ICAM (from IFI27^+^), THY1 (from IFI27^+^), and GP1BA (from BC-ES). Despite leaning toward the IFI27^+^ lineage, BC-AS2 does not exhibit pronounced NOTCH signaling. IFI27^+^ cells do express NOTCH but its level remains relatively low, suggesting that NOTCH is crucial for BCs to maintain normal epithelial homeostasis. The loss or downregulation of NOTCH may drive BCs toward a pro-inflammatory phenotype, potentially exacerbating epithelial aging. For ECM-receptor interactions (Figure 2H), the SC subpopulation receives a large share of ECM-related input signals (Input Pattern 1 includes COLLAGEN, LAMININ, FN1, THBS, and TENASCIN). The BC-AS2 and IFI27^+^ subpopulations primarily exhibit outgoing ECM signals (Output Pattern 1 includes COLLAGEN, FN1, THBS, TENASCIN, and RELN). These data suggest that BC-AS2 and IFI27^+^ cells are critical sources of ECM components during skin aging, contributing to basement membrane and extracellular matrix remodeling. For secretory signals (Figure 2J), the IFI27^+^ subpopulation also shows markedly enhanced outgoing and incoming signals, including immune-related pathways (e.g., IL1, IL2, IL4, IL6, and TGFB), underscoring its inflammatory nature. BC-ES cells have an elevated output of signals such as EGF, IGF, IGFBP, BMP, EDN, PLAU, and FLT3, as well as an increased reception of VISFATIN (from IFI27^+^), PERIOSTIN (from BC-AS2), and COMPLEMENT (from IFI27^+^). The BC-AS2 subpopulation shows a clear trend toward stronger output of PTPR and PERIOSTIN signals, whereas the BC-AS1 subpopulation receives more WNT and FASLG signals, reflecting its inclination toward the SC lineage. Collectively, these findings suggest that disruptions in BC secretory functions—especially aberrations in EGF, IGF, and IGFBP signaling, combined with heightened PTPR and PERIOSTIN pathways—may induce a pro-inflammatory reprogramming of BCs, thereby promoting their senescence and ultimately accelerating the skin aging process.

### 2.3. Keratinocyte Cell–Cell Interaction Features in Young and Aged Skin

From the perspective of cell–cell contacts (Figure 3), we observed notable changes in the signaling pathways along the developmental trajectory from the BC-ES to BC-AS2 and IFI27^+^ subpopulations. First, the loss of CDH1 signaling was identified in this direction, which is tightly linked to epithelial differentiation. Its disappearance also implies a loss of epithelial polarity, potentially facilitating abnormal processes such as epithelial-mesenchymal transition. Second, JAG1-NOTCH1/NOTCH2/NOTCH3, a critical pathway in skin epithelial differentiation, was lost as BC-ES progressed toward BC-AS2. Meanwhile, DLL1-NOTCH1/NOTCH2/NOTCH3 was lost as BC-AS2 developed into IFI27^+^ cells, and DLL1-NOTCH2 was also lost during the BC-ES to BC-AS2 transition. Furthermore, these signals remained absent when considering the reverse influences of the IFI27^+^ subpopulation on both BC-ES and BC-AS2 cells. Third, the loss of EFNB-EPHB signaling emerged as another defining feature of the BC-ES to IFI27^+^ trajectory. EFNB-EPHB participates in cell-cell recognition, modulates cell migration and tissue boundary formation, and governs cell positioning during development. Its absence may therefore disrupt normal epithelial proliferation and differentiation, ultimately altering cell fate. From the standpoint of the IFI27^+^ subpopulation, we noted a marked upregulation of the SELE-CD44 pathway, which likely exerts varying degrees of influence on all subpopulations. Additionally, IFI27^+^ cells, together with BC-AS2 cells in the same developmental branch, showed enhanced ICAM1 signaling (ITGAX/ITGAM/LTGAL-ITGB2), again affecting all subpopulations to different extents. The IFI27^+^ subset also exhibited other significant immunomodulatory signals, such as the upregulation of multiple HLA family members. These observations collectively suggest that IFI27^+^ cells may have acquired immunoregulatory functions, shifting away from a normal epithelial fate and instead adopting an aberrant, pro-inflammatory phenotype.

### 2.4. ECM Receptor Characteristics of Keratinocyte Subpopulations in Young and Aged Skin

From the perspective of ECM receptor features (Figure 4), both the BC-AS2 and IFI27^+^ subpopulations exhibit significant alterations in collagen–integrin and collagen–CD44 signaling. These changes include enhanced COL–ITGA1/ITGA2/ITGB1 interactions, along with strengthened COL–CD44 direct-binding signals. Specifically, the IFI27^+^ subpopulation shows upregulation of COL1A1, COL1A2, COL4A1, COL4A2, and COL6A1–COL6A3, whereas BC-AS2 primarily involves COL1A1, COL1A2, COL6A1, and COL6A2. These observations underscore the pivotal contribution of specialized cell populations to ECM remodeling in aged skin, especially through collagen–integrin and CD44-mediated pathways. The IFI27^+^ subpopulation’s more comprehensive collagen expression profile suggests a heightened capacity for matrix remodeling and cell migration, while BC-AS2—although mostly modulated by type I and VI collagens—also displays augmented integrin signaling (ITGA1/2 + ITGB1), indicating a distinct functional role in cell–matrix interactions. Such differential ECM receptor alterations not only reflect functional divergence in microenvironmental adaptation but also highlight potential therapeutic targets for modulating these specialized signaling pathways.

### 2.5. Secreted Signaling Features Among Keratinocyte Subpopulations in Young and Aged Skin

In-depth analysis of the secreted signaling heatmaps revealed a complex network of regulatory pathways (Figure 5). Notably, WNT3A–FZD8/FZD6/LRP5/LRP6 interactions exhibited strong signaling activity between BC-ES cells and both BC-AS1 and the SC subpopulations. In contrast, VEGFA–VEGFR1/VEGFR2 and members of the TGF-β superfamily were preferentially upregulated in the IFI27^+^ subpopulation. Regarding inflammatory mediators, IL1, IL2, IL6, and various TNF superfamily members—as well as chemokine axes such as CCL21–CCR7 and CCL5–CCR5—were significantly enhanced in IFI27^+^ cells. Additionally, the evolving BC-AS1 and BC-AS2 subpopulations, along with IFI27^+^ cells, showed active growth factor signaling (e.g., FGF family) and heightened matrix-related molecular interactions. Collectively, these findings illuminate a multilayered regulatory framework governing cell fate determination and point toward novel therapeutic strategies that target specific signaling pathways. At the same time, this intricate signaling network underscores the need to account for compensatory mechanisms and dynamic inter-subpopulation signaling when developing anti-aging interventions, thereby emphasizing the importance of a comprehensive, science-driven approach to mitigating skin aging.

### 2.6. Transcriptomic Characteristics of Trajectory-Specific Genes in Keratinocyte Subpopulations of Young and Aged Skin

Based on the developmental trajectories, we extracted the characteristic genes from the three developmental stages—ES, AS1, and AS2—and evaluated their expression levels using skin epithelial transcriptomic data. For the ES signature genes (Figure 6A), there were relatively minimal differences in expression between young and aged individuals under normal conditions, apart from a small subset of CXCL family members (*CXCL2* and *CXCL3*, which were higher in young skin) and genes such as *DMKN* and *CCL21*, which were more abundantly expressed in aged skin. In general, these ES genes remained comparatively stable at the transcriptomic level. However, after exposure to ultraviolet (UV) radiation, both young and aged epidermal cells showed significantly increased expression of these ES genes. Strikingly, aged skin displayed a more pronounced inflammatory response under UV stress, evidenced by notably high expression of *IFI27, CD74, CTS3, S100A4, LY6E, HLA-DPA1, VIM,* and *HSPA8*. These findings indicate that, although ES genes exhibit cross-age stability under normal physiological conditions, they demonstrate a marked stress response under exogenous challenges (e.g., UV exposure)—a response that is more strongly inflammatory in aged skin. The upregulation of immune-related molecules (e.g., *IFI27*, *CD74*, and *HLA-DPA1*), stress proteins (HSPA8), and cytoskeleton remodeling genes (VIM and S100A4) underscores a heightened reactivity and impaired homeostatic control in aged epithelium. This age-dependent divergence in stress response likely reflects microenvironmental reprogramming that increases epidermal vulnerability to external insults, providing a molecular explanation for the heightened susceptibility to skin disorders in the elderly and suggesting new targets for preventive and therapeutic interventions.

Regarding the AS1 branch (i.e., genes directed toward SC differentiation; Figure 6B), the transcriptomic profiles of epidermal cells from young and aged individuals displayed both shared features and distinct patterns. Overall, young epidermal cells showed elevated expression of genes involved in transcriptional regulation (*EIF3A*, *EIF4B*, *ZNF770*, and *DDX46*), cellular homeostasis (*TMEM154*, TMEM256, VAMP2, and *KPNB1*), and metabolic regulation (*IGBP1*, *LSM5*, and *PPTC7*). These changes suggest an active transcriptional and metabolic reprogramming process encompassing protein synthesis, RNA metabolism, membrane trafficking, and cellular communication. Additionally, the upregulation of stress response genes (*OSTC*, *GAPG*, *HSPA8*, and *RORA*) indicates that these reprogramming events may represent adaptive mechanisms to cope with physiological or pathological challenges. In aged epidermal cells, the AS1 gene signature primarily involved core biological processes related to immune regulation (*STAT3* and *IL1RN*), metabolic modulation (*APOE* and *LDLR*), and signal transduction (*PKD1* and *RHOV*), alongside genes influencing cell proliferation (*CCND1*), differentiation (*GATA3*), and stress response (*HEBP2* and *WASP2*). Following UV exposure, young individuals exhibited heightened expression of metabolism-related genes (*FERMT1*, *POP7*, *PSMA10*, and *UBR4*) and transcriptional regulators (e.g., *EIF*, *RIOK3*, and *EHF*), whereas aged epidermal cells displayed a pronounced increase in inflammatory genes, such as *IFNGR*, *ARF5*, *TRIM29*, *DSC1*, *JUP*, and *LAMP1*. These differential expression patterns suggest that each cell population may activate distinct functional modules in response to various stimuli and underscore the metabolic and stress-related shifts characteristic of skin aging.

For the AS2 branch (i.e., cells developing toward the IFI27^+^ subpopulation; Figure 6C), young epidermal cells exhibited only a limited set of characteristic genes, including *CXCL12*, *EIF5A*, *CXCL3*, and *TIMP2*. In contrast, aged epidermal cells showed pronounced immune- and metabolism-related gene signatures such as *HLA-DRA*, *CCL21*, *CXCL1*, *APOE*, and *AQP1*. Upon UV irradiation, both young and aged epidermal cells displayed substantially increased expression of these inflammation-related genes, with *CD74*, *CTS3*, and *IFI27* notably elevated in both groups. These observations indicate that, during aging, epidermal cells progressively acquire a pro-inflammatory phenotype that can be further exacerbated by exogenous stressors such as UV radiation. The sustained upregulation of *IFI27* in particular suggests that it may serve as a critical molecular node linking aging, UV-induced damage, and inflammatory responses. Specifically, the baseline activation of immune-related genes (*HLA-DRA*, *CCL21*, and *CXCL1*) in older individuals—an “inflammaging” signature—becomes further amplified upon UV exposure, and the marked elevation of *IFI27* highlights its potential role in mediating stress responses and inflammatory cascades during skin aging. This age-dependent increase in inflammation susceptibility and the activation of IFI27-associated pathways provide mechanistic insight into why older skin exhibits heightened sensitivity to environmental insults.

### 2.7. Genetic Loci Associated with Skin Aging and Their Influence on Developmental Trajectory Genes

To investigate the genetic underpinnings of facial aging, we analyzed GWAS data specifically focused on facial skin aging. We first identified significant loci that exert an influence on facial aging (Figure 7A), which revealed notable signals across multiple chromosomes, with chromosomes 2, 3, 6, and 9 being particularly prominent. We then matched the genes proximal to these significant loci and intersected them with the trajectory-specific signature genes described earlier. This approach uncovered several significant loci associated with multiple developmental trajectory genes (Figure 7B), including SH3YL1, which harbors multiple SNP sites. We next performed enrichment analyses on these neighboring genes. The results for biological processes (Figure 7C) indicated that these genes are primarily involved in extrinsic growth and intrinsic apoptotic pathways, as well as protease protection, hydroperoxide activity, and ruffle organization/assembly. Other enriched processes encompass purine nucleoside and triphosphate metabolic pathways, as well as functional modules related to the inclusion of body-domain-containing proteins and proteasomal protein catabolism. Collectively, these findings reflect the broad range of key biological functions activated during cellular stress responses. Furthermore, the KEGG pathway analysis reiterated that these proximal genes contribute to antigen processing and endoplasmic-related molecular signaling pathways (Figure 7D). Taken together, our data underscore the complex, polygenic regulation characteristic of facial aging, particularly highlighting significant loci on chromosomes 2, 3, 6, and 9 in spatial proximity to developmental trajectory genes. Functional enrichment analyses of these associated genes reveal a regulatory network encompassing cell growth control, apoptosis, protein degradation, oxidative stress response, and immune processing. Notably, the enrichment of antigen processing pathways and endoplasmic-related signaling further emphasizes the central role of immune–inflammatory responses in skin aging. This multilayered functional convergence provides a systematic molecular framework for understanding the genetic underpinnings of facial aging and indicates potential avenues for targeted anti-aging interventions.

### 2.8. Hypothetical Molecular Model of Skin Aging

Based on the integrative evidence from single-cell analyses, transcriptomics, and genetic loci, we propose the following mechanistic hypothesis for skin aging (Figure 8). Under normal physiological conditions, WNT and FASLG signaling pathways precisely regulate the proliferation, differentiation, and fate determination of BCs and SCs, thereby preserving skin tissue homeostasis and renewal. However, dysregulation of key pathways, such as PTPR and PERIOSTIN, triggers a shift toward a pro-inflammatory microenvironment, leading to aberrant basal cell function and the disruption of tissue homeostasis. This, in turn, sets off a cascade of complex molecular events, including reactive oxygen species (ROS) accumulation and oxidative stress damage, progressive DNA lesion build-up, abnormal transcriptional and mRNA expression profiles, imbalanced activation of pro-inflammatory mediators and cytokine networks, reprogramming of cellular energy metabolism, and disruptions in epidermal differentiation. These interacting and mutually reinforcing molecular processes ultimately drive the transition of skin tissues from a healthy to an aged state, characterized by disorganized epidermal structure, compromised basement membrane integrity, degradation of dermal collagen and elastin, and reduced barrier function—hallmarks of skin aging. Elucidating the multifaceted interplay among these signaling pathways and molecular mechanisms provides a systematic theoretical framework for understanding the essence of skin aging. It also highlights critical directions for developing targeted anti-aging interventions and personalized skin care strategies.

## 3. Discussion

This study employed publicly available single-cell sequencing data, transcriptomics, and facial aging-related GWAS data to comprehensively investigate keratinocyte subpopulations and their developmental trajectories during skin aging. First, through clustering analyses of young and aged skin keratinocytes, we identified three main subpopulations: basal cells (BCs), spinous cells (SCs), and IFI27^+^ keratinocytes. This categorization not only aligns with classical knowledge of skin physiology at the phenotypic level but also reveals a potentially critical role for a novel subpopulation (IFI27^+^) in maintaining or accelerating skin aging. Second, using single-cell pseudotime analysis (Monocle 2), we identified three distinct differentiation branches—ES (early state), AS1 (bias toward SC differentiation), and AS2 (bias toward IFI27^+^ cells). The findings suggest that basal cells may diverge along different branches during aging, heading either toward spinous differentiation (AS1) or an inflammatory state (AS2), the latter being particularly associated with the high expression of inflammation-related genes. Whereas traditional studies often focus on aging features in the basal or spinous layers alone, our work delves more deeply into the multi-level differences exhibited by distinct keratinocyte subpopulations, notably the significant increase in IFI27^+^ cells in aged skin and their pro-inflammatory properties. This is especially evident in analyses of cell–cell communication, ECM-receptor interactions, and secretory signal networks: IFI27^+^ cells display more active outgoing and incoming inflammatory and immunomodulatory signals, suggesting that they may not merely respond passively to the microenvironment but may also actively accelerate aging and promote chronic inflammation. Furthermore, by subdividing basal cells into BC-ES, BC-AS1, and BC-AS2 based on pseudotime, we were able to elucidate how BC-ES cells transition in different directions and how they interact with neighboring subpopulations (SC and IFI27^+^) via multiple signaling pathways—such as NOTCH, PTPR, and PERIOSTIN—which exhibit distinct characteristics along each branch. In addition, transcriptome data provided supplementary evidence to corroborate the single-cell discoveries. Under normal conditions, young and aged skin showed certain differences in the expression of ES-, AS1-, and AS2-related genes; following UV exposure, however, these differences became more pronounced, especially reflected in stronger inflammatory responses in aged skin. Lastly, by integrating GWAS data with differentially expressed genes from the single-cell trajectories, we found that significant loci on chromosomes 2, 3, 6, and 9 were spatially correlated with multiple key biological pathways (including antigen processing, oxidative stress, and apoptosis), underscoring the polygenic, multi-signal regulatory features of skin aging. Overall, this study offers novel and comprehensive insights into keratinocyte subpopulation diversity, developmental trajectories, and signaling networks during skin aging, laying important groundwork for future molecular diagnostic and therapeutic strategies.

Skin aging is a complex biological process driven by both intrinsic and extrinsic factors, centered on the dynamic imbalance of different cell types and their microenvironments [21,22,23]. Our research shows that BCs, traditionally regarded as the source of skin renewal, exhibit distinct differentiation trajectories in aging: one branch moves toward the SC lineage (AS1), maintaining some degree of keratinization, while the other progresses toward an inflammatory path (AS2) closely associated with IFI27^+^ cells. Moreover, from the perspective of single-cell communication, the signaling interactions between BCs and the surrounding cells become increasingly aberrant with age—evidenced by the loss of NOTCH, the disappearance of CDH1, and the overactivation of PTPR and PERIOSTIN pathways. These disruptions not only appear as consequences of aging but may also exacerbate or amplify pro-inflammatory microenvironments in aging skin. Among multiple signaling cascades, WNT and FASLG have long been recognized as essential in controlling BC proliferation and stem cell homeostasis [24,25,26,27]. In young skin, these pathways help balance the interplay between BCs and SCs, maintaining a healthy epidermal renewal cycle. Once perturbed by internal or external stressors (e.g., UV radiation, oxidative stress, or genetic mutations), these signaling pathways can become dysregulated, prompting BCs toward an inflammatory phenotype and increasing IFI27^+^ cell proportions. The IFI27^+^ cells, in turn, secrete various pro-inflammatory factors (such as IL1, IL6, and TNF-α family members), establishing a positive feedback loop that intensifies inflammation and erodes homeostasis. Additionally, our analysis of ECM receptor features and secretory signals suggests that abnormal activation of collagen–integrin/CD44 signaling not only reflects the disordered remodeling of the skin matrix but also drives abnormal cell adhesion, migration, and morphological changes, hastening the loss of structural and functional integrity of keratinocytes [28,29,30].

It is worth noting that cross-talk between fibroblasts, immune cells, endothelial cells, and basal cells is critically important in skin aging. Though this study primarily focuses on keratinocytes, the gene enrichment and signaling analyses revealed that many immune-related genes (e.g., HLA family, *CD74*, and *ICAM1*) were upregulated in both IFI27^+^ cells and BC-AS2 subpopulations, implying that the basement membrane and dermal microenvironment also undergo significant alterations. These findings highlight that skin aging does not merely involve the decline in a single cell type but is instead a multi-cell, multi-signal, and multi-pathway process. By combining single-cell sequencing and transcriptomics, we have gained new insights for dissecting each cell subpopulation and pathway with greater granularity, thereby illuminating the cooperative or antagonistic factors that drive aging. In short, the multidimensional mechanism of skin aging is reflected in the differentiation and inflammatory changes in various subpopulations, alongside the complex interplay of cross-cell and cross-pathway regulations. This networked shift across multiple dimensions underlies the gradual and often irreversible nature of skin aging. A deep understanding of this network not only helps identify the most promising therapeutic targets or cell subpopulations for intervention but also informs future preventive and therapeutic strategies.

By utilizing GWAS data specifically related to facial skin aging, we identified significant loci on chromosomes 2, 3, 6, and 9 that closely correlate with the aging process. Furthermore, intersecting these SNPs with the differentially expressed genes from our single-cell trajectory analyses revealed potential connections between genes (e.g., *SH3YL1*) and key signals in skin aging. These loci may directly or indirectly affect various aspects of keratinocyte function, such as proliferation and differentiation, antioxidative capacity, and immune regulatory pathways, thus influencing both the onset and progression of skin aging at the molecular level. Enrichment analyses indicated that many of these nearby genes are closely involved in extrinsic growth, intrinsic apoptotic pathways, protein catabolism, oxidative stress, and immune responses. Previous studies on skin aging have often emphasized environmental factors [31,32,33,34] (e.g., UV exposure, pollution, and stress); however, our findings from the genetic perspective provide strong evidence for a multi-gene regulatory component in skin aging. With the accumulation of DNA damage and a reduction in repair efficiency during aging, specific SNPs or gene mutations could lead to accelerated or more severe aging phenotypes, affecting collagen synthesis, fibroblast activity, and inflammatory responses. Importantly, the enrichment of antigen processing pathways (e.g., the HLA family) and endoplasmic reticulum stress-related signals underscores the critical roles of immune mechanisms and protein-folding quality control in skin aging—a pattern consistent with the heightened immune-related gene expression observed in IFI27^+^ and BC-AS2 subpopulations, suggesting an intrinsic coupling between genetic factors and immune–inflammatory networks. Elucidating these multifaceted genetic associations not only enhances our understanding of the molecular basis of skin aging but also opens new avenues for personalized medicine and targeted interventions. For instance, if we can clinically detect an individual’s high-risk SNPs related to key areas of chromosomes 2, 3, 6, and 9, we may intervene earlier through anti-inflammatory approaches, ROS reduction, or enhancement of protein homeostasis. Future efforts to integrate epigenetic studies (DNA methylation, histone modifications, and non-coding RNAs) with GWAS data will clarify how gene–environment interactions drive the transition of skin cells toward aging phenotypes at a deeper molecular level. Taken together, this work underscores a complex interplay among genes, the environment, developmental trajectories, and intercellular communication—laying a robust foundation for understanding the causes of skin aging and establishing potential therapeutic targets.

The multifaceted analysis of skin aging in this study provides several clinical and industrial applications. First, in early diagnosis and molecular subtyping, the marked increase in IFI27^+^ cells and the heightened expression of pro-inflammatory genes (e.g., *CD74* and *CTS3*) offer promising biomarkers for evaluating the extent of skin aging and underlying inflammation. Dermatologists and cosmetic practitioners may incorporate these markers into skin biopsies or RNA-based assays for more accurate assessments of aging status and inflammation risk, enabling more tailored skin care and treatment plans. Moreover, the emphasis placed on certain genes (e.g., *SH3YL1* and the HLA family) by the GWAS analysis suggests that genetic background testing could help identify high-risk individuals earlier, facilitating timely intervention via lifestyle modifications or targeted treatments.

Second, multiple signaling pathways (NOTCH, WNT, FASLG, PTPR, and PERIOSTIN) identified in this study appear pivotal in regulating both basal cell fate and inflammatory processes. Developing agents or inhibitors that specifically target these pathways may represent a promising new approach for decelerating skin aging. For example, inhibiting the excessive expression of PTPR and PERIOSTIN could mitigate the pro-inflammatory environment, and modulating WNT or NOTCH signaling might preserve normal differentiation and the repair capacities of basal cells. Additionally, the link between collagen–integrin/CD44 interactions and basement membrane remodeling indicates that scientifically formulated collagen products or integrin pathway modulators may help maintain skin barrier integrity and tissue structure.

Third, our findings highlight a range of molecular targets with potential for cosmetic product development. The shared roles of IFI27^+^ cells and BC-AS2 in inflammation, immune regulation, and ECM remodeling suggest that therapeutic strategies aiming to downregulate pro-inflammatory factors or block certain collagen degradation pathways might yield more effective anti-aging outcomes. Further, by focusing on genes that become markedly upregulated under UV exposure in aged skin (e.g., *IFI27*, *VIM*, and *CD74*), skincare formulations could incorporate active ingredients that attenuate ROS and dampen inflammatory cascades, thereby mitigating photo-induced aging.

Finally, it is important to note individual variability and compensatory mechanisms. Skin aging is not merely the simple depletion of a single pathway but rather a dynamic imbalance among multiple signals. Relying solely on interventions for one pathway or gene may be insufficient due to biological redundancy or compensatory shifts in other pathways. A holistic, integrative approach—merging multi-omics data with precision medicine while considering each individual’s genetic background and environmental exposures—is essential for truly effective prevention and therapy.

Despite the comprehensive analysis presented here, integrating multiple omics datasets to explore the skin aging process, several limitations remain that warrant attention. First, public database samples often suffer from inconsistent sources, differing sample sizes, and various sequencing platforms, leading to potential data heterogeneity or bias. This variability can affect the precision of clustering results and add noise to integrative analyses such as cross-referencing GWAS and single-cell differentially expressed genes [35,36]. Future research could employ larger, more homogeneous datasets and merge multi-center data to improve reliability and generalizability.

Second, while single-cell sequencing and transcriptomic analyses reveal dynamic cellular and molecular states, these findings lack robust functional validation. For instance, we observed a clear link between the expansion of IFI27^+^ cells and increased inflammation in aged skin but causality remains to be confirmed through in vitro or in vivo models. Further experimental studies—such as targeted knockout or overexpression of critical pathways [37,38] (NOTCH, PTPR, and PERIOSTIN)—are necessary to elucidate their precise roles in promoting inflammation and accelerating skin aging.

Third, skin aging is inherently spatiotemporal, and our current analysis largely relies on transcriptomic and single-cell data from epidermal compartments. Deeper insights into fibroblasts, immune cells, vascular endothelial cells, and neuroendocrine cells in the dermis and how they coordinate with the epidermis during aging are still lacking. Moreover, our analysis does not adequately address the critical influence melanocytes have on keratinocyte behavior throughout the aging process, despite the well-established communication between these cell types via melanin transfer and paracrine signaling. In reality, aging often involves immune cell infiltration, changes in extracellular matrix composition, modifications to dermal–epidermal coupling, and alterations in melanocyte–keratinocyte interactions that affect photoprotection, pigmentation, and oxidative stress responses. Applying spatial transcriptomics or advanced multiplex fluorescence in situ hybridization (mFISH) in conjunction with 3D skin models would provide more comprehensive spatiotemporal resolution for understanding the interplay among skin layers and cell types throughout aging [39], including the crucial melanocyte–keratinocyte dynamics.

Lastly, while our study provides valuable insights into keratinocyte heterogeneity through transcriptomic analysis, a significant limitation is the lack of integration between the genetic variants (GWAS data) and epigenomic analyses. Though we identified aging-associated subpopulations and their transcriptional signatures, we could not determine how specific SNPs affect epigenetic modifications, transcription factor binding, or 3D chromatin architecture that might drive this heterogeneity. A growing body of evidence suggests that epigenetic regulation—DNA methylation, histone modifications, non-coding RNAs—may even surpass direct genetic variation in establishing and maintaining the cellular heterogeneity we observed during aging. Our current approach cannot capture how epigenetic landscapes diverge among different keratinocyte subpopulations, potentially causing functionally identical cells to adopt specialized states with age.

Looking ahead, the multi-dimensional, multi-subpopulation, multi-pathway insights offered by this study point to several promising research directions. First, integrating multiple omics is imperative. Single-cell transcriptomics alone can delineate cellular heterogeneity but falls short of capturing detailed protein, metabolic, epigenetic, and spatial information. Combining spatial transcriptomics, proteomics, and metabolomics could paint a more complete molecular and cellular portrait of skin aging, helping to pinpoint the most critical nodes in the aging process.

Second, deeper investigations into extracellular environment regulation and skin microecology are warranted [40]. The skin surface is home to numerous microorganisms and interacts with various environmental factors (light, humidity, and pollutants), collectively forming a complex ecosystem with the epidermal and dermal immune barriers. Aging may involve not only internal cellular changes but also microbiota shifts, altered barrier permeability, and other environmental stress responses. A multi-angle approach could clarify how external and microbial factors coordinate with cell fate determination to shape aging phenotypes.

Third, our study’s emphasis on IFI27^+^ cells, BC-AS2 differentiation pathways, and core routes (PTPR and PERIOSTIN) provides new leads for drug discovery or functional skincare development. Future work can systematically screen for compounds that inhibit inflammation, enhance basal cell proliferation, or promote ECM reconstruction. In vitro experiments, followed by clinical trials, will be crucial for translating these insights into safe and effective anti-aging solutions.

Finally, stem cell and tissue engineering approaches may further validate and apply these findings. Constructing artificial 3D skin models or organoids simulating natural aging and external stressors would allow for the real-time tracking of cell differentiation, inflammation, and matrix remodeling. Such models could also be coupled with CRISPR/Cas9 gene editing for more precise functional tests. In addition to enhancing our fundamental understanding, these methods could pave the way for innovative regeneration medicine targeting severe skin damage or age-related deterioration.

## 4. Materials and Methods

### 4.1. Single-Cell Sequencing Data Acquisition and Keratinocyte Subgroup Identification

The target dataset (GSE130973) was obtained from the GEO database [41]. Based on the original author’s cell type annotations, 2323 keratinocytes were extracted for subsequent analysis. Data normalization was performed using the LogNormalize method with a scale factor of 10,000 in the Seurat package (version 4.4.0) [42]. Then, the highly variable genes were identified using the variance stabilizing transformation (VST) method, selecting 2000 features. After scaling the data with the cell cycle scores (S.Score and G2M.Score) regressed out, principal component analysis (PCA) was performed using the selected variable features, followed by Harmony integration to correct for batch effects. The top 30 principal components were used for non-linear dimensionality reduction (UMAP), and clustering analysis was performed using the Louvain algorithm (resolution = 0.2). Subgroup annotation was based on known keratinocyte marker gene expression patterns: basal cells (BCs) using KRT5 and KRT14 as characteristic genes, and squamous cells (SCs) using KRT1 and KRT10 as characteristic genes [10]. The FindAllMarkers function (wilcox.test, min.pct = 0.25, logfc.threshold = 0.25) was used to identify subgroup-specific expressed genes.

### 4.2. Developmental Trajectory Analysis of Skin Keratinocytes

Cell differentiation trajectories were constructed using Monocle 2 [43] (version 2.18.0). Dimensionality reduction was first performed based on differentially expressed genes (*q*-value < 0.01), and branch structures were constructed using the DDRTree algorithm [44]. The orderCells function was used to determine cell developmental order and cells were ordered by pseudotime. Cells were grouped by trajectory states, and the proportions of different cell subgroups in each state were calculated. FindAllMarkers was used to compare state-specific gene differences [42]. The GO enrichment analysis (clusterProfiler (version 4.6.2) package, *p*.adjust < 0.05) and KEGG pathway analysis (*p*.adjust < 0.05) were performed on these genes [45,46].

### 4.3. Cell–Cell Communication Network Analysis

CellChat [47] (version 1.1.3) was used to analyze cell–cell interactions. The parameters were set as follows: raw.use = TRUE; population.size.min = 10. The analysis included ligand-receptor-mediated cell communication, extracellular matrix (ECM) interactions, and secretory-factor-mediated signaling pathways. The netVisual_aggregate function was used to generate circle plots of cell–cell communication intensity. The netVisual_heatmap and netVisual_river were used to display ligand-receptor pair signaling patterns. The netVisual_bubble was used to display detailed cell pair interactions in specific signaling pathways.

### 4.4. Transcriptome Data Validation

The GSE67098 [48] dataset was obtained, which contained transcriptome data from young (<35 years) and aged (>60 years) skin samples from sun-exposed and sun-protected areas. This dataset specifically included Caucasian subjects to control for pigmentation factors, with paired samples collected via 4-millimeter-diameter punch biopsies from the outer forearm or lateral epicanthus (sun-exposed areas) and upper inner arm (sun-protected areas). Differentially expressed genes with percentage (PCT) differences > 0.25 were selected from single-cell data and their mean expression values were calculated for the transcriptome data. The pheatmap package (version 1.0.12) was used to generate heatmaps showing the expression patterns of these genes in young and aged skin.

### 4.5. Genetic Variation and Skin Aging Association Analysis

GWAS data for facial skin aging (ID: ukb-b-2148; sample size: 423,999; number of SNPs: 9,851,867; population: European) was obtained from the OpenGWAS database (https://gwas.mrcieu.ac.uk/datasets/ukb-b-2148/, accessed on 5 January 2025) [49,50]. Significant SNPs were filtered using the following criteria: *p* < 5 × 10^−8^; window size for linkage disequilibrium (LD): 1000 kb; r^2^ = 0.01 [51]. The vautils package (https://github.com/oyhel/vautils/tree/master, accessed on 5 January 2025) was used to obtain neighboring genes of significant SNPs. These genes were intersected with differentially expressed genes from the trajectory analysis to identify potentially functionally related genes. GO and KEGG enrichment analyses (*p*.adjust < 0.05) were performed using clusterProfiler (version 4.6.2) to compare biological function similarities with trajectory state characteristic genes [52].

## 5. Conclusions

Through an integrated multi-omics approach combining single-cell transcriptomics, bulk RNA sequencing, and genome-wide association studies, this research has elucidated a critical bifurcation in the fate of BCs during epidermal aging, where BCs either follow canonical differentiation into spinous cells or transition toward an inflammatory IFI27^+^ phenotype that amplifies age-associated inflammation and orchestrates extracellular matrix remodeling. These findings reveal that maintaining BC homeostasis—specifically by preventing their inflammatory reprogramming—represents a pivotal axis for mitigating cutaneous senescence, identifying promising therapeutic targets including the PTPR, PERIOSTIN, and NOTCH signaling pathways. Collectively, the evidence underscores the polygenic and pleiotropic mechanisms driving skin aging and suggests that strategic interventions focused on preserving BC stability may constitute an effective approach for attenuating age-related dermatological changes.

## Figures and Tables

**Figure 1 ijms-26-02617-f001:**
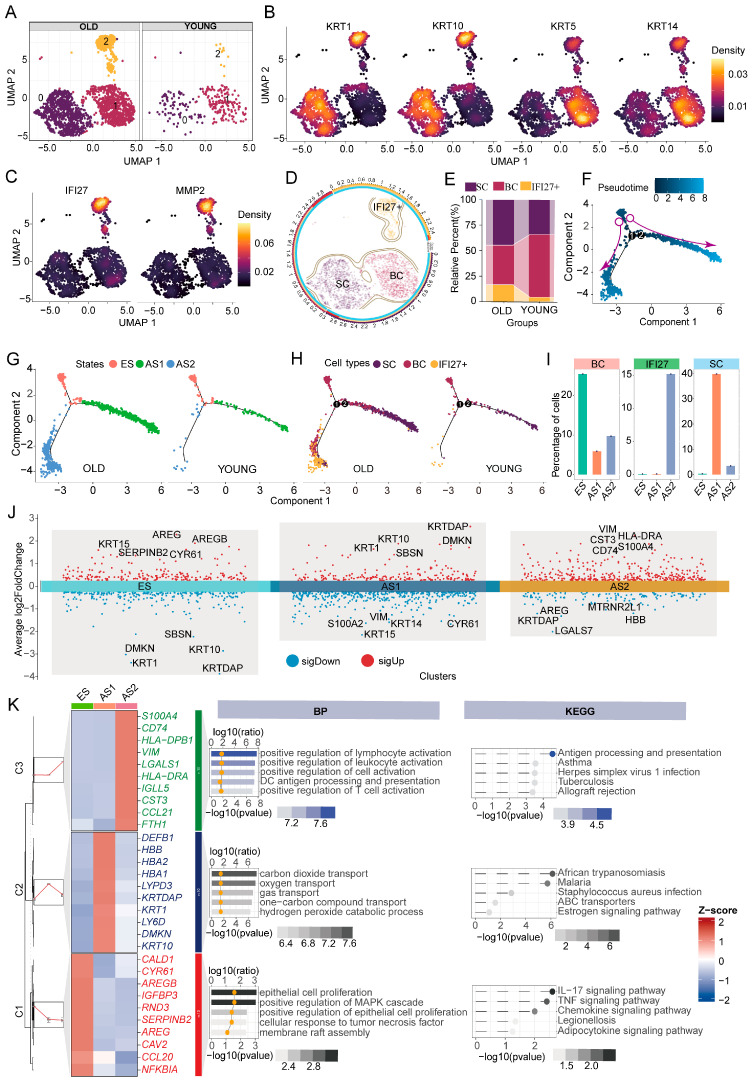
Keratinocyte distribution patterns and developmental trajectories in young and aged facial skin: (**A**) Clustering analysis of keratinocytes from young and aged skin samples. (**B**) Distribution patterns of characteristic epithelial genes (*KRT1*, *KRT10*, *KRT5*, *KRT14*) across keratinocyte populations. (**C**) Cellular distribution patterns of *IFI27* and *MMP2* expression. (**D**) Overview of keratinocyte subpopulations. (**E**) Comparative analysis of epithelial subpopulation proportions between young and aged individuals. (**F**) Developmental trajectory of keratinocytes, with nodes ① and ② indicating developmental checkpoints and arrows showing the developmental direction. The color gradient from dark blue to light blue represents progression from early to late developmental stages. (**G**) Cell distribution patterns across different developmental states in young and aged individuals, where ES represents the early stage and AS1-2 represents two distinct advanced directions. (**H**) Distribution patterns of epithelial subpopulations along developmental trajectories in young versus aged individuals. (**I**) Differential proportions of developmental states across keratinocyte subpopulations. (**J**) Differential gene expression patterns across developmental states, with red indicating high expression and blue indicating low expression. (**K**) Comprehensive analysis showing the following: (**left**) heatmap of top 10 marker genes across different developmental states; (**middle**) enriched biological processes represented as horizontal bar charts; and (**right**) enriched KEGG pathways displayed as lollipop plots.

**Figure 2 ijms-26-02617-f002:**
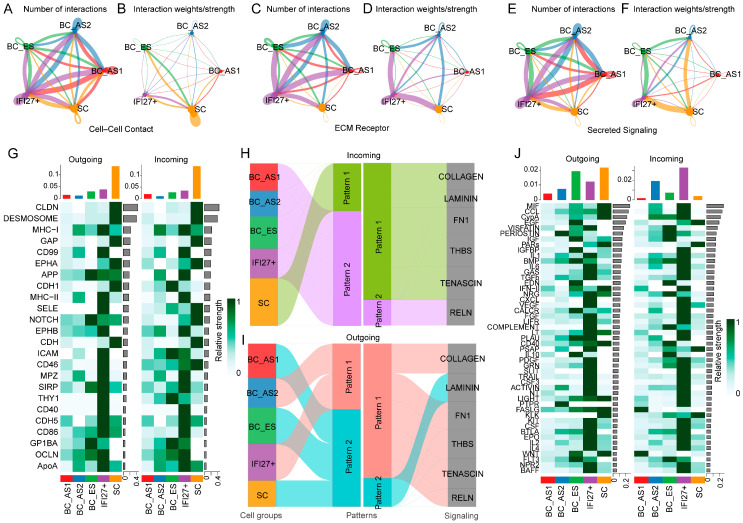
Cell–cell interaction characteristics between keratinocyte subpopulations in young and aged skin: (**A**) Quantification of cell–cell interactions between epithelial subpopulations across developmental stages. (**B**) Interaction strength analysis between epithelial subpopulations across developmental stages. (**C**) Quantification of ECM–receptor interactions between epithelial subpopulations across developmental stages. (**D**) ECM–receptor interaction strength analysis across developmental stages. (**E**) Quantification of secretory signaling between epithelial subpopulations across developmental stages. (**F**) Secretory signaling strength analysis across developmental stages. (**G**) Heatmap depicting changes in input and output cell–cell signaling between epithelial subpopulations across developmental stages. (**H**,**I**) River plots showing changes in ECM-receptor input (**H**) and output (**I**) signaling, and (**J**) Heatmap showing changes in secretory signaling input and output between epithelial subpopulations across developmental stages. BC, basal cell; SC, spinous cell; ES, early state; AS, advanced state; IFI27^+^, cells highly expressing gene IFI27.

**Figure 3 ijms-26-02617-f003:**
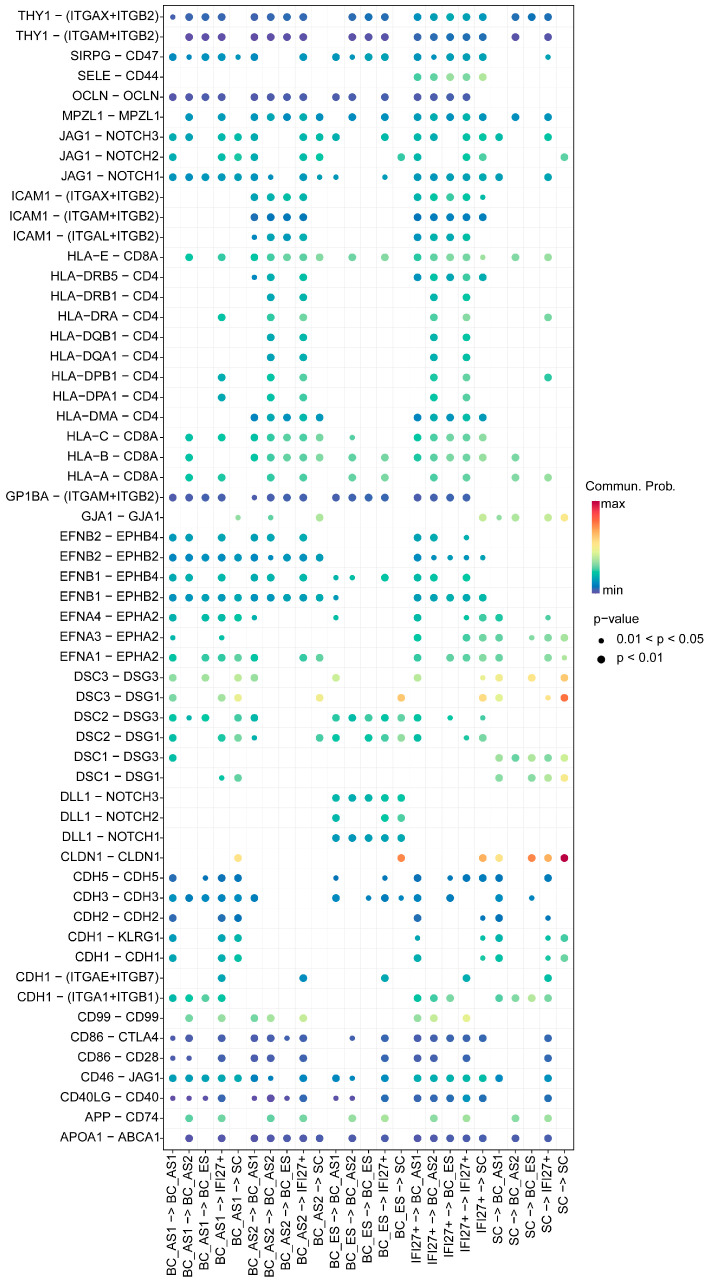
Ligand-receptor characteristics of cell–cell interactions between skin epithelial subpopulations across developmental stages. The bubble plot displays interaction intensities represented by a color gradient from dark blue (weak) to dark red (strong). Bubble sizes indicate statistical significance, with larger bubbles representing smaller *p*-values and thus higher significance. The x-axis shows cell–cell interaction directions, while the y-axis displays ligand-receptor pairs. BC, basal cell; SC, spinous cell; ES, early state; AS, advanced state; IFI27^+^, cells highly expressing gene IFI27.

**Figure 4 ijms-26-02617-f004:**
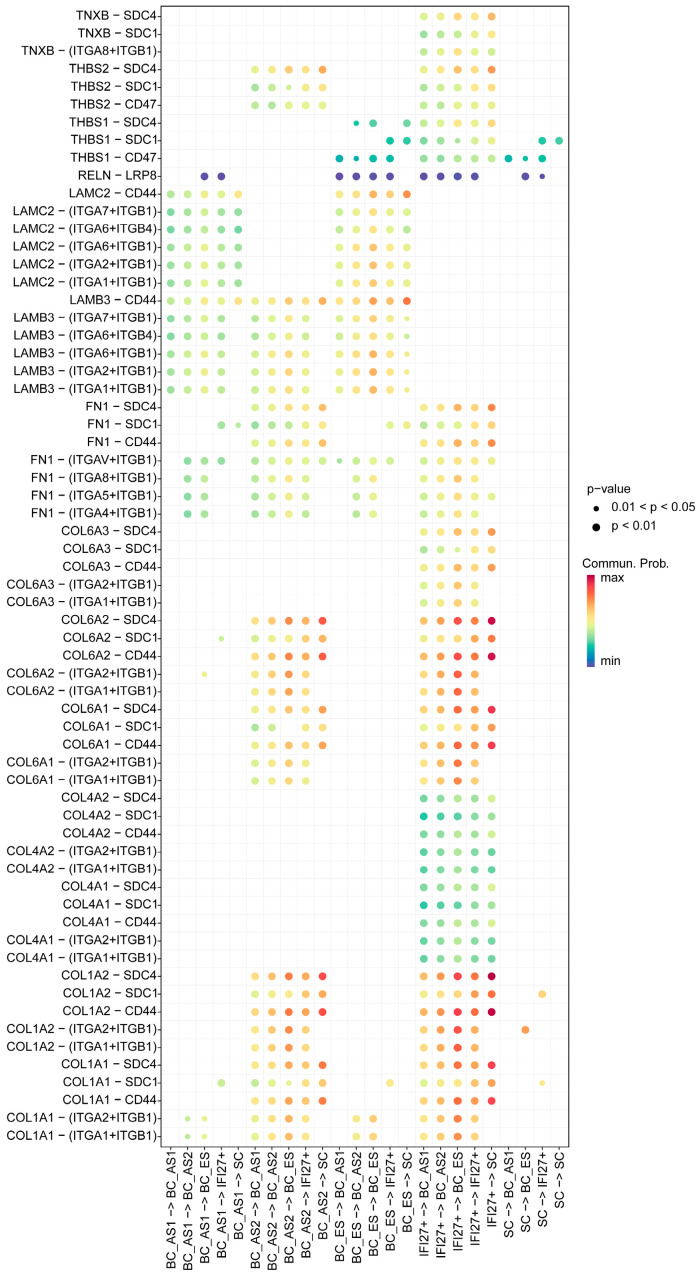
Ligand-receptor characteristics of extracellular matrix receptor between skin epithelial subpopulations across developmental stages. The bubble plot displays interaction intensities represented by a color gradient from dark blue (weak) to dark red (strong). Bubble sizes indicate statistical significance, with larger bubbles representing smaller *p*-values and thus higher significance. The x-axis shows cell–cell interaction directions, while the y-axis displays ligand-receptor pairs. BC, basal cell; SC, spinous cell; ES, early state; AS, advanced state; IFI27^+^, cells highly expressing gene IFI27.

**Figure 5 ijms-26-02617-f005:**
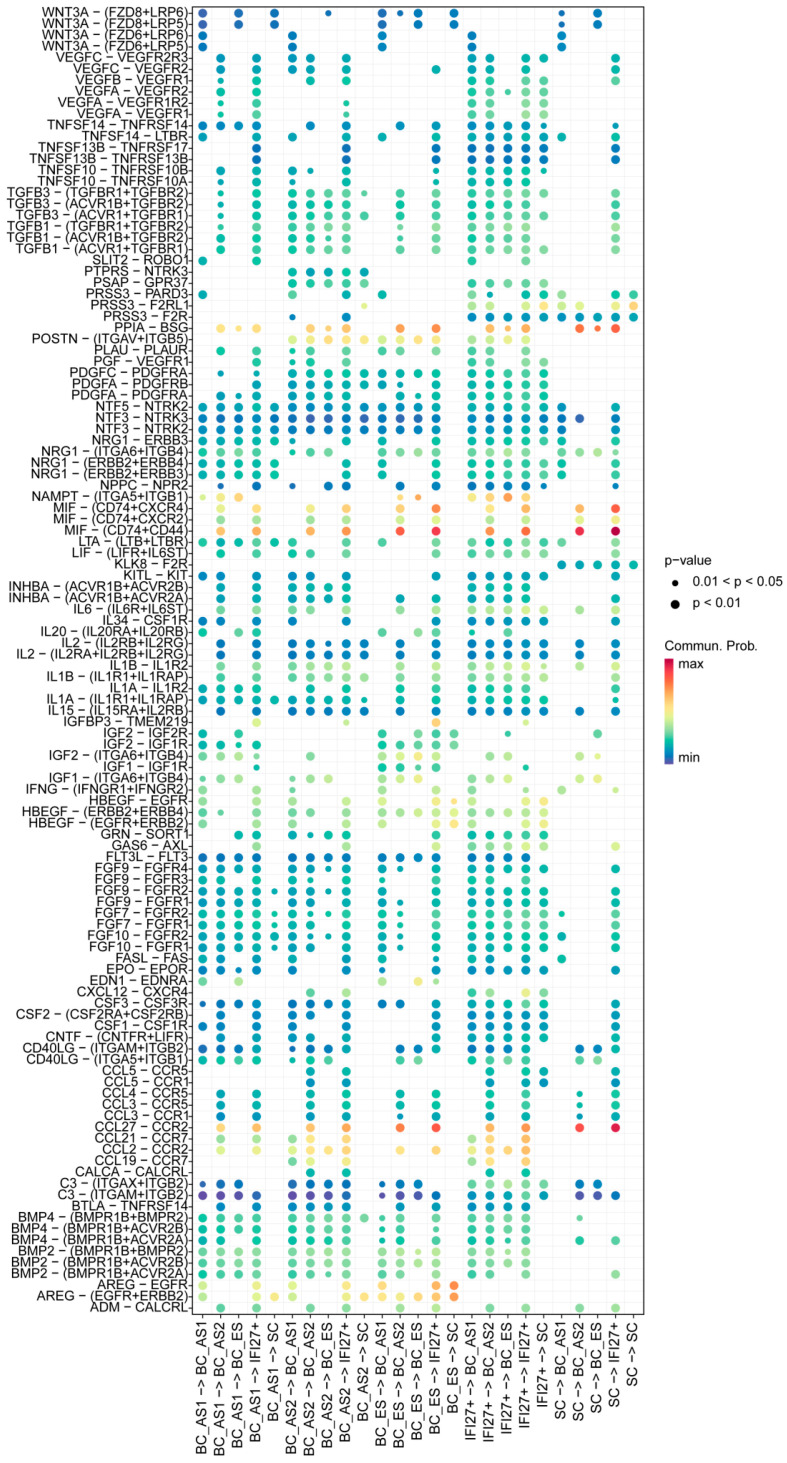
Ligand-receptor characteristics of secreted signaling between skin epithelial subpopulations across developmental stages. The bubble plot displays interaction intensities represented by a color gradient from dark blue (weak) to dark red (strong). Bubble sizes indicate statistical significance, with larger bubbles representing smaller *p*-values and thus higher significance. The x-axis shows cell–cell interaction directions, while the y-axis displays ligand-receptor pairs. BC, basal cell; SC, spinous cell; ES, early state; AS, advanced state; IFI27^+^, cells highly expressing gene *IFI27*.

**Figure 6 ijms-26-02617-f006:**
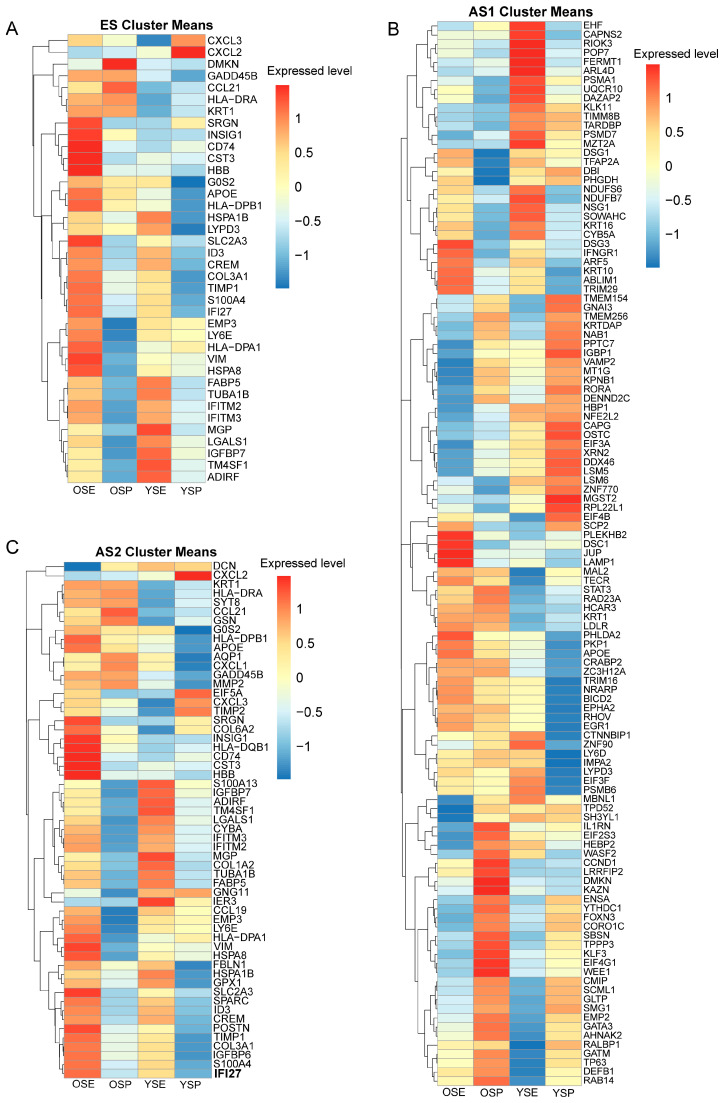
Heatmap of differentially expressed genes across developmental stages in bulk RNA sequencing of skin epithelial tissue: (**A**) early stage genes; (**B**) genes involved in basal cell (BC) to spinous cell (SC) cell differentiation stages; (**C**) genes involved in BC to IFI27^+^ cells. OSE, skin tissue from aged individuals with history of UV exposure; OSP, skin tissue from aged individuals with history of UV protection; YSE, skin tissue from young individuals with history of UV exposure; YSP, skin tissue from young individuals with history of UV protection.

**Figure 7 ijms-26-02617-f007:**
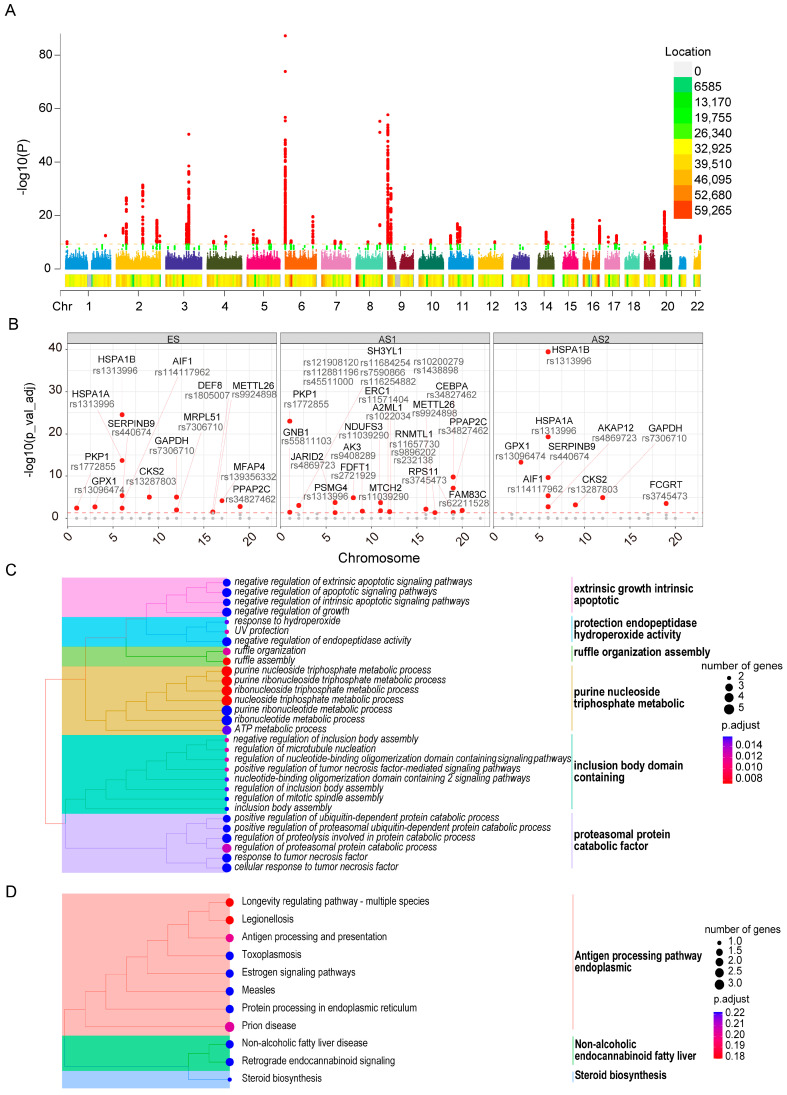
GWAS analysis reveals genetic loci distribution of characteristic genes in skin keratinocyte subpopulations across different stages: (**A**) distribution patterns of significant genetic loci; (**B**) proximity relationships between significant genetic loci and stage-specific keratinocyte developmental genes; (**C**) biological processes enriched in genes adjacent to significant loci; (**D**) KEGG pathways enriched in genes adjacent to significant loci.

**Figure 8 ijms-26-02617-f008:**
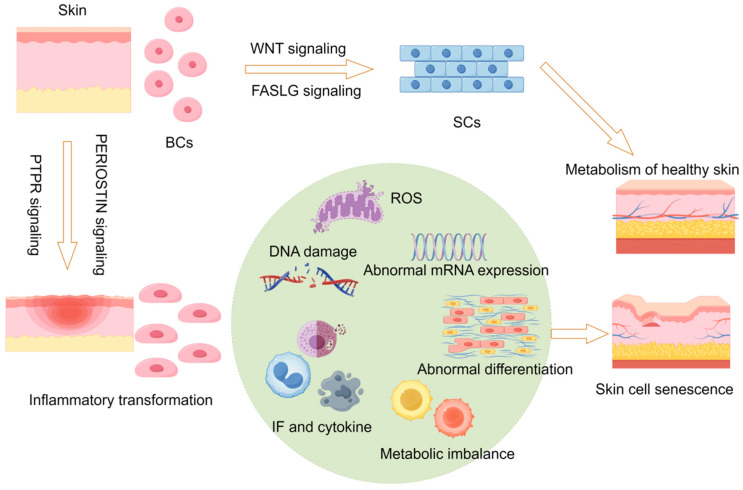
Hypothetical molecular model of skin aging. BCs, basal cells; SCs, spinous cells; IF, interferon; ROS, reactive oxygen species.

## Data Availability

The datasets generated and/or analyzed during the current study are available in the IEU OpenGWAS project (https://gwas.mrcieu.ac.uk/, accessed on 10 January 2025) and GEO database (https://www.ncbi.nlm.nih.gov/gds, accessed on 20 December 2024).

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
