# Peer review of "Inflammatory Transformation of Skin Basal Cells as a Key Driver of Cutaneous Aging"

_ijms, 2025, doi:10.3390/ijms26062617_

Round 1
Reviewer 1 Report
Comments and Suggestions for Authors
Review Report on
Inflammatory Transformation of Skin Basal Cells as a Key Driver of Cutaneous Aging
Important queries:
- The abstract part should be expanded with the contemporary results.
- In the introduction part, the citations are required to be added for the last paragraph, especially for the present results?
- Need citation for the single-cell RNA-sequencing (scRNA-seq) data?
- What is the full form of GWAS?
- Single-cell RNA-sequencing (scRNA-seq) analysis of Epithelial Cell Distribution Patterns and Developmental Trajectories in Young and Aged Facial Skin results in Figure 1, the overall gene expressions were well established with the research data.
- According to the literature the results matched perfection where in Section 3.2. Fig. 2 shows cell-cell interactions over epithelial Cell Subpopulation Interactions in Young and Aged Skin where the interactions of epithelial cell subpopulations in young and aged skin differ in a number of ways, including cell type proportions, signaling, and immune cell composition. These differences contribute to the impaired wound healing and reduced barrier function characteristic of aged skin. However, in aged skin, basal keratinocytes' relative abundance decreases, while spinous keratinocytes' abundance increases.
- Sec.3.4 and Figure 4, Ligand-Receptor Characteristics of Extracellular Matrix Receptor Between Skin Epithelial Subpopulations Across Developmental Stages. The bubble plot displays interaction intensities represented by a color gradient from dark blue (weak) to dark red (strong). Bubble sizes indicate statistical significance, with larger bubbles representing smaller p-values and thus higher significance. The x-axis shows cell-cell interaction directions, while the y-axis displays ligand-receptor pairs. BC basal cells; SC, spinous cells; ES, early state; AS, advanced state; IFI27+, cells highly expressed gene IFI27. In this section, the authors should cite more references to provide practical evidence.
- Sec.3.8 and Fig .8, Hypothetical Molecular Model of Skin Aging showed excellent results between young and aged skin through cell metabolism.
- The conclusion part should be expanded on par with research results.
Although the overall research data is well established and the manuscript contained few English corrections, which should be minimized.
Result: Minor revision.

Reviewer 2 Report
Comments and Suggestions for Authors
Having read the manuscript I have the following comments.
- In this manuscript you state epithelial to refer to keratinocytes which is incorrect, you will find ~10% of this layer consists of melanocytes which produce melanin which is transferred to keratinocytes. Please replace epithelial cells to that of keratinocytes.
- Why is no reference made anywhere to the influence melanocytes have on the behaviour of the keratinocytes.
- In ref 28 the GSE67098 dataset used in this study, what is known about the irradiated skin samples found in this study? What was the skin type of people used in this study what was the UV exposure they were exposed to? What influence did the melanocytes and age as well as UV exposure have in these different skin tissues?
- What changes the keratinocyte population would be seen in the skin from a person who has Fitzpatrick type 1 skin to someone with type IV or V skin? It is not clear from your study which skin type is this study based on, can this be clarified.
Reviewer 3 Report
Comments and Suggestions for Authors
What techniques were used to investigate the heterogeneity of epithelial cells in ageing skin?
Was it not clear from the text the relationship between the loss of NOTCH signaling and skin aging?
How does the remodeling of the extracellular matrix influence skin aging? and how could external factors, such as U|V rays, influence this study?
What basis for therapeutic products do the authors believe could be viable for slowing down skin ageing?
Do the authors believe that future biomarkers and methodologies could be developed based on these results? And what are the limitations of this study? What else needs to be done? It would be good to create a topioc of future perspectives.
Round 2
Reviewer 2 Report
Comments and Suggestions for Authors
I would like to thank the authors for making the changes to the manuscript, and making it clearer that the study refers to changes in the development and differentiation of keratinocytes found in the stratum basale. I have no further questions or comments on the manuscript.